# Bound Electron Enhanced Radiosensitisation of Nimorazole upon Charge Transfer

**DOI:** 10.3390/molecules27134134

**Published:** 2022-06-28

**Authors:** Sarvesh Kumar, Islem Ben Chouikha, Boutheïna Kerkeni, Gustavo García, Paulo Limão-Vieira

**Affiliations:** 1Atomic and Molecular Collisions Laboratory, CEFITEC, Department of Physics, Universidade NOVA de Lisboa, Campus de Caparica, 2829-516 Caparica, Portugal; s.kumar@campus.fct.unl.pt; 2Département de Physique, LPMC, Faculté des Sciences de Tunis, Université de Tunis el Manar, Tunis 2092, Tunisia; islem.benchouikha@fst.utm.tn; 3ISAMM, Université de La Manouba, La Manouba 2010, Tunisia; 4Instituto de Física Fundamental, Consejo Superior de Investigaciones Científicas, Serrano 113-bis, 28006 Madrid, Spain; g.garcia@csic.es

**Keywords:** nimorazole, electron transfer, radiosensitizer, TOF mass spectrum, energy loss spectrum

## Abstract

This novel work reports nimorazole (NIMO) radiosensitizer reduction upon electron transfer in collisions with neutral potassium (K) atoms in the lab frame energy range of 10–400 eV. The negative ions formed in this energy range were time-of-flight mass analyzed and branching ratios were obtained. Assignment of different anions showed that more than 80% was due to the formation of the non-dissociated parent anion NIMO^•−^ at 226 u and nitrogen dioxide anion NO_2_^−^ at 46 u. The rich fragmentation pattern revealed that significant collision induced the decomposition of the 4-nitroimidazole ring, as well as other complex internal reactions within the temporary negative ion formed after electron transfer to neutral NIMO. Other fragment anions were only responsible for less than 20% of the total ion yield. Additional information on the electronic state spectroscopy of nimorazole was obtained by recording a K^+^ energy loss spectrum in the forward scattering direction (*θ* ≈ 0°), allowing us to determine the most accessible electronic states within the temporary negative ion. Quantum chemical calculations on the electronic structure of NIMO in the presence of a potassium atom were performed to help assign the most significant lowest unoccupied molecular orbitals participating in the collision process. Electron transfer was shown to be a relevant process for nimorazole radiosensitisation through efficient and prevalent non-dissociated parent anion formation.

## 1. Introduction

Nimorazole (NIMO≡C_9_H_14_N_4_O_3_) is a chemical compound that has been widely used as an efficient hypoxic (deficient in oxygen) cell radiosensitizer in head and neck squamous carcinoma due to its low toxicity [1]. In its composition, NIMO contains a 4-nitroimidazole and a morpholine rings bonded by their nitrogen atoms (N1) through a –H_2_C–CH_2_– side group (Figure 1).

NIMO’s ability to scavenge low-energy electrons was recently shown to be very efficient in producing a non-dissociated parent anion with a considerable large cross-section of ~3 × 10^−18^ m^2^ (at ~0 eV) [2,3], while dissociative electron attachment was shown to be a minor reaction channel which is suppressed upon hydration [2]. Although the detailed mechanisms by which radiosensitisation operates are still unknown, it is believed that radiosensitisers are chemical compounds subject to redox reactions inside the hypoxic cells [4], and in case of nitroimidazoles, the ring facilitates reduction through reactive anion radicals’ formation [5,6,7]. Nitroimidazole radiosensitisers have been thoroughly investigated by experimental methods on low-energy electron interactions [8,9,10,11,12] and together with nimorazole probed by electrospray ionization mass spectrometry [13,14,15]. While associative electron attachment may contribute to NIMO’s radiosensitising effect, within the biological environment, electron transfer processes (redox reactions) may prevail and so these may seem more appropriate to describe the underlying molecular mechanisms of such chemical compounds and their role as radiosensitisers. Thus, more than a decade, we initiated an investigation methodology to explore key selected radiosensitisers by charge transfer in atom-molecule collision experiments, viz. halogenated uracils [16] and, more recently, nitroimidazoles [17,18]. In the former case, the decomposition yields significant ring breaking with appreciable NCO^−^ intensities (in 5-FU) and halogen anion formation (in 6-ClU), thus compromising the integrity of viral RNA where the neutral molecules bind. For the latter, the collision induced dissociation results on the formation of several radicals, with particular attention to NO^•^ and ^•^OH. These radicals act as an indirect DNA damage agent triggered by ionising radiation, while nitric oxide has been noted for sensitising hypoxic solid tumours in radiotherapy and chemotherapy treatments [19]. Another relevant aspect pertains to the role of site and bond-selectivity in electron transfer processes to DNA/RNA nucleotides yielding bond cleavage [20,21], which have been shown to be pivotal in controlling chemical reactions. As recently shown [17,18], such control of a given bond breaking may have relevance to tailor chemical control for different applications such as tumor radiation therapy through nitroimidazole-based radiosensitisation.

Here, we present a novel and comprehensive investigation of NIMO negative ion formation in electron transfer processes in a wide collision energy range, combining experimental and state-of-the art theoretical methods. Section 2 deals with the results and Section 3 presents the discussion, which includes a complete description of the electronic states probed by the experimental method and supported by quantum chemical calculations. Section 4 is devoted to a short description of the experimental setup and the computational details of the calculations that are used to interpret the experimental data. Finally, Section 5 includes a brief summary of our major findings and details some conclusions drawn from the present joint experimental and theoretical investigations.

## 2. Results

Negative ion yields of NIMO have been obtained as a function of the collision energy from 10 to 400 eV in the lab frame (7.7–310 eV in the center-of-mass frame), with their proposed assignments in Table 1. Note that the observed relative isotopic yields are in accordance with the expected natural abundances for the cyanide anion (CN^−^) at 26 u (100%) and 27 u (1.44%), the nitrogen dioxide anion (NO_2_^−^) at 46 u (100%), 47 u (0.44%) and 48 (0.4%), and the non-dissociated parent anion NIMO^•−^ at 226 u (100%), 227 u (11.5%), 228 u (1.2%) and 229 u (~0.1%). In Table 1, we also list the anions formed in (dissociative) electron attachment experiments [2,3], noting that in the present investigation, the fragmentation pattern is much richer. This was certainly not unexpected given the different collision dynamics in electron transfer to a free electron process. The two most intense signals are assigned to the non-dissociated parent anion NIMO^•−^ at 226 u and the nitrogen dioxide anion NO_2_^−^ at 46 u (see Figure 2), amounting, on average, to more than 90% of the total anion yield. The rich fragmentation pattern, which is more enhanced at higher collision energies (above 40 eV in Figure 2), reveals significant collision-induced decomposition of the 4-nitroimidazole ring, as well as other complex internal reactions within the temporary negative ion formed after electron transfer to neutral NIMO. Yet, these fragment anions are only responsible for less than ~10% of the total ion yield. The branching ratios (BRs) for the most intense anions from NIMO are shown in Figure 3, where a strong energy dependence is discernible. These have been obtained from the integrated time-of-flight mass peak divided by the total anion yield at a given collision energy. This means that, although the non-dissociated parent anion yield in the TOF mass spectra is lower than NO_2_^−^ (Figure 2), its integrated signal comprises 226 u and its isotopes, yielding a branching ratio value that is dominant over all the other fragment anions across the entire collision energy range investigated.

Figure 4 depicts the post-collision potassium cation (K^+^) energy loss spectrum at 157 eV in the center-of-mass frame (205 eV in the lab frame) in the forward scattering direction (*θ* ≈ 0°). The energy loss data have been smoothed and properly fitted with Gaussian functions to decompose the spectrum, with vertical values given by the different peaks’ maxima.

The vertical electron affinities and assignments of the most typical molecular orbitals are shown in Table 2. The energy loss needed to access an electronic state is given by:ΔE = IE(K) − EA(I_max_)(1)
with IE(K) the ionization energy of the potassium atom (4.34 eV [22]) and EA(I_max_) the target’s molecule electron affinity for that state [23]. Figure 4 shows a main feature with a maximum intensity (I_max_) centered at (7.21 ± 0.03) eV, yielding a vertical electron affinity of (−2.87 ± 0.03) eV, in good accord with the dissociative electron attachment resonance at ~2.97 eV [3] (see discussion further down).

The electronic structure of NIMO in the presence of a potassium atom has also been investigated by theoretical methods (see Section 4.2), where the shape and energy of the different molecular orbitals (MOs) have been obtained up to 10 eV (see Appendix A). The information gained from the lowest unoccupied molecular orbitals (LUMOs) is critical to assign the different electronic states attained in the electron transfer process. The geometry of nimorazole was optimized at the M06-2X/6-311++g(d,p) level of theory. Upon collision with the potassium atom, the system K + NIMO was also optimized at the M06-2X/6-311++g(d,p) level of theory. The optimized geometry led to a distance of ≈5.1 Å between K and O atoms from the 4-nitroimidazole ring (see Figure 5).

Note that in K + NIMO collisions, the major fragment anion yield is assigned to NO_2_^−^, so the calculations were performed with the potassium atom pointing at the –NO_2_ end group from the 4-nitroimidazole ring, as shown in Figure 5.

## 3. Discussion

In potassium (K)-molecule (M) collision experiments yielding ion-pair formation, the collision complex is formed by the oxidized and the reduced species, i.e., K^+^ and M^–#^. While K^+^ is still in the vicinity of the metastable parent anion (M^–#^), a strong Coulomb interaction can delay autodetachment long enough to allow the stabilization of the negative ion via intramolecular energy redistribution through the different available degrees of freedom, resulting either in a stable parent anion or in fragmentation channels (via direct or statistical dissociation). Relative to dissociative electron attachment, in electron transfer experiments, one can attain identical anions’ formation with different yields, as well as other exit channels, yielding different anionic species. This has been reported several times in the past (see, e.g., [24,25,26,27,28,29,30,31,32] and references therein) and is solely due to the different collision dynamics dictating the underlying molecular mechanism responsible for negative ion formation [33]. Briefly, in atom-molecule collisions, in the vicinity of the crossing between the ionic and covalent potential energy curves governing the electron transfer process, electrons follow adiabatically the nuclear motion. The endoergicity (ΔE) of the process yielding ion-pair is given by the difference between the ionization energy of the electron donor and the electron affinity of the target molecule [23,34] (see Equation (1)). Since the collision energy is typically set above the threshold of ion-pair formation, the metastable parent anion is formed with an excess of internal energy.

Anion yields from time-of-flight (TOF) mass spectra as a function of the K + NIMO collision energy have been obtained from 10 to 400 eV in the lab frame (7.7–310 eV in the center-of-mass frame), where a rich fragmentation pattern is observed (assignments in Table 1) mainly at higher energies (see Figure 2). The major ion signal is due to the non-dissociated molecular anion NIMO^•−^ followed by NO_2_^−^ formation. Based on the present experimental data and theoretical calculations, we now discuss in detail the electronic state spectroscopy of the most representative anions formed in collisions of neutral potassium atoms with neutral nimorazole molecules, viz. NIMO^•−^, NO_2_^−^, 4-nitroimidazole and morpholine rings fragmentation, CN^−^, CNO^−^ and (NIMO–NO)^−^.

### 3.1. Parent Anion

The TOF mass spectra recorded in a wide collision energy range shows that the non-dissociated parent anion contributes to more than 60% of the total ion yield for E_CM_ < 10 eV, while above 100 eV, its branching ratio is ~80% (see Figure 3). This is a strong indication that NIMO is extremely efficient in scavenging an electron being transferred from the potassium donor. The reaction that yields the parent anion is given by:K + NIMO → (K^+^ NIMO^•–#^) → K^+^ + (NIMO^•–^)^#^ → K^+^ + NIMO^•−^(2)
with (NIMO^•−^)^#^ the temporary negative ion (TNI) formed with an excess of internal energy. Formation of the non-dissociated parent anion may strongly compete with either autodetachment or molecular dissociation, the latter clearly visible from the fragmentation pattern obtained from 10 to 50 eV (Figure 3). However, owing to the different internal degrees of freedom, efficient intramolecular energy redistribution may occur, leaving the parent anion intact within the 80 μs time detection of the present experiment.

The calculated highest occupied molecular orbitals in the presence of a potassium atom show a delocalized spin density over the entire molecule (HOMO-1) with a prevalent character around the 4-nitroimidazole and morpholine rings (HOMO) (Figure 6). This is also the case in the bare NIMO molecule (Appendix A). As far as the lowest unoccupied molecular orbitals are concerned (see Appendix A), in the presence of a potassium atom, a significant delocalization over the molecule is observed, which can be responsible for NIMO^•−^ formation (e.g., LUMO + 50 in Figure 6). As the collision energy is increased, one would expect more energy deposited in the NIMO molecule, thus increasing its fragmentation yield and decreasing the non-dissociated parent anion signal. However, NIMO^•−^ becomes dominant for the entire collision energy range, albeit with a modest increase in fragment anion yields from 7.7 up to ~50 eV in the center-of-mass (Figure 6). This tendency of the NIMO^•−^ yield pertains to the role of the electronic structure, where a more enhanced delocalized spin density over all molecule is discernible from the high energy LUMOs (Appendix A). Above ~70 eV, the parent anion BR, as well as the other fragment anions, are insensitive to the collision energy, the former contributing to 80% of the total anion yield. This means that the electron transfer process may occur when the potassium atom flies apart from the NIMO molecule, leaving the TNI with an excess of internal energy that is not efficiently channeled into the fragmentation channels. In such an energy range, the collision time is faster, <23 fs, and the efficient Coulomb stabilization of the collision complex (K^+^ NIMO^•–#^) no longer efficiently holds.

Within the scope of electron transfer, we noted a set of molecular systems where a non-dissociated parent anion is formed in, e.g., highly symmetric molecules such as hexachlorobenzene C_6_Cl_6_ [35] (six identical C–Cl bonds) and in cyclic chemical compounds such as nitroimidazoles [18] (strong competition between the ring and the—NO_2_ end). Other molecular targets where strong evidence of effective bond breaking and rearrangement are operative, further to the excess energy deposited within the TNI, are the cases of thymine C_5_H_6_N_2_O_2_ and uracil C_4_H_4_N_2_O_2_ [36], nitromethane CH_3_NO_2_ [37] and acetic acid CH_3_COOH [38], just to mention a few.

It was interesting to note that electron attachment studies have revealed that NIMO^•–^ is by far the dominant anion with a very narrow feature at ~0 eV, with an absolute cross-section value of ~3 × 10^−18^ m^2^ [2]. The shape of the molecular orbitals in Appendix A (LUMO to LUMO + 7) clearly show the predominant character over the entire molecule to attach an extra electron. Although nimorazole is of low symmetry, its 84 different vibrational degrees of freedom have been suggested to provide an effective means of dissipating the excess energy, leading to a stable NIMO^•−^ anion within the detection time window of a few hundred microseconds [2].

The post-collision potassium cation (K^+^) energy loss spectrum at 157 eV in the center-of-mass frame (205 eV in the lab frame) in the forward scattering direction (*θ* ≈ 0°) is shown in Figure 4. The lowest energy-loss feature peaks at (4.74 ± 0.07) eV and results in an electron affinity of (−0.40 ± 0.07) eV (Table 2). Taking the C–NO_2_ bond dissociation energy in 4-nitroimidazole to be 3.19 eV [39], the NO_2_ electron affinity of (2.2730 ± 0.0050) eV [22], and a calculated adiabatic electron affinity of 1.31 eV at the M062x/6-311 + G(d,p) level of theory [6], the asymptotic limit of (NIMO-NO_2_) + NO_2_^–^ can be obtained at 2.227 eV, i.e., 0.917 eV above the ground state of the neutral molecule. This means that the 4.74 eV feature does not lead to bond excision but rather NIMO^•–^ formation.

### 3.2. NO_2_^−^ Formation

The nitrogen dioxide anion BR is the second most intense in K + NIMO collisions, contributing to more than 20% of the total ion yield for E_CM_ < 50 eV, while above 80 eV, its value accounts, on average, for 15% (see Figure 3). Electron promotion to NIMO via the 4-nitroimidazole ring may yield the nitrogen dioxide anion, given the high electron affinity of NO_2_. In the unimolecular decomposition of the TNI, the reaction involves cleavage of the C–NO_2_ bond from the nitroimidazole ring, with the extra charge on the NO_2_ radical, and can be given by:K + NIMO → (K^+^ NIMO^•−#^) → K^+^ + (NIMO^•−^)^#^ → K^+^ + NO_2_^−^ + (NIMO–NO_2_)(3)
yet with no evidence of the complementary reaction leading to the loss of a neutral NO_2_ and formation of (NIMO–NO_2_)^−^, as is observed in potassium collisions with 2-and 4(5)-nitroimidazole [18].

As far as the lowest unoccupied molecular orbitals are concerned, a significant delocalization over the molecule together with considerable σ*(N–C) antibonding character between the 4-nitroimidazole ring and the –H_2_C–CH_2_– side group (LUMO + 50) are responsible for NIMO^–^ and NO_2_^–^ formation, respectively. From the BRs in Figure 3, we note that the relative intensity of NO_2_^–^ has a maximum at ~30 eV, decreasing from 30 up to 70 eV, and its yield remains somehow constant (~15%) with increasing collision energy.

We obtained the asymptotic limit of (NIMO-NO_2_) + NO_2_^−^ at ~0.92 eV above the ground state of the neutral NIMO molecule and determined that the first energy loss feature at 4.74 eV (Figure 4) is mainly due to the non-dissociated parent anion formation (see Section 3.1). However, the DEA data of Meißner et al. [3] shows a weak NO_2_^−^ feature at 0.41 eV, which was assigned to the excitation of the symmetric stretching vibration mode of NO_2_ with an energy of 0.163 eV (1318 cm^−1^) in the neutral ground and 0.159 eV (1284 cm^−1^) in the negative ion states [22]. Thus, a possible operative mechanism can be related to the role of vibrationally excited neutral molecules, even with modest population at 374 K heating temperature of NIMO. Notwithstanding, a more plausible explanation can reside in the nature of the multidimensional potential energy surfaces of NIMO in respect to the reaction coordinate leading to C–NO_2_ bond excision. Meißner et al. [2] reported a weak NO_2_^−^ signal at electron energies close to 0 eV due to an exothermic reaction with ΔG = −0.26 eV corresponding to the release of C_3_H_3_N_2_ + C_6_H_11_NO.

The energy loss feature at (6.11 ± 0.04) eV (Figure 4) is closely related to accessing an electronic state at 1.77 eV. The calculated MOs (LUMO + 60 in Appendix A) show a $\sigma_{CN}^{*}$ antibonding character in the 4-nitroimidazole ring, with the extra electron sitting on NO_2_, although we should not discard the possibility of other fragment anions stemming from the nitroimidazole ring being formed, viz. CN^–^. Yet, and according to the DEA data [3], other close lying resonance at 1.93 eV can be responsible for the cyano radical anion formation, which is clearly visible from the LUMO + 15 in Appendix A. The next energy loss feature, peaking at (7.21 ± 0.03) eV, yields a vertical electron affinity of (−2.87 ± 0.03) eV, which can be assigned to the DEA resonance at 2.97 eV resulting in NO_2_^−^ formation [3]. The calculated MOs show that LUMO + 70 (Appendix A) at 7.01 eV depicts a strong $\sigma_{CN}^{*}$ antibonding character in the 4-nitroimidazole ring. A close inspection of Figure 4 reveals that this energy loss feature has a threshold of ~5 eV, which is in excellent accord, within the energy resolution of the K^+^ energy loss data, with the obtained value of 0.92 eV if we now add the potassium ionization energy of 4.34 eV [22].

### 3.3. Other Fragment Anions

Figure 3 shows that fragment anions stemming from the 4-nitroimidazole ring, the morpholine ring and other anions (CN^−^ and CNO^−^) in K + NIMO collisions contribute to less than 15% of the total ion yield for E_CM_~50 eV, while above this energy they account only for just ~10%. Such ion yields decrease at higher energies, which is related to fast collisions where the strong Coulomb stabilization within the collision complex is no longer efficient enough to allow the opening of different accessible dissociation channels.

At 10 eV center-of-mass frame (Figure 3), CN^−^ and CNO^−^ contribute to ~20% of the total anion yield. The neutral species are known to be pseudohalogens considering their electron affinities, EA(CN) = 3.86 eV and EA(CNO) = 3.61 eV [22], exceeding those of the halogen atoms. It is interesting to note that such high electron affinities do not mean higher yields of the fragment anions, which has also been reported in electron transfer experiments to nitroimidazoles [17,18], pyrimidine [30], adenine [21,40], thymine and uracil [36,41]. CN^−^ and CNO^−^ are mostly formed from the decomposition of the 4-nitroimidazole ring upon electron transfer to NIMO. Such is the assertion of the similarity found in the resonance shapes and positions of these anions, together with NO_2_ in DEA experiments [3]. The reactions yielding such anions’ formation involve several bonds being broken, the former from the loss of an O atom from NO_2_ and excision of two C–N bonds within the (4-nitro)imidazole ring, whereas the latter through the loss of an N atom from the NO_2_ or by multiple bond breaking directly from the imidazole ring. A close inspection of the molecular orbitals in Appendix A (see Appendix A) shows a modest antibonding character within the 4-nitroimidazole ring involving the C–N bonds, which may be related to the lowest ionic yields obtained in the different TOF mass spectra as a function of the collision energy (see, e.g., from LUMO + 50 to LUMO + 100).

The loss of a neutral NO^•^ (196 u) belongs to the set of trace anions with a branching ratio < 5%, and can be formed through the following reaction:K + NIMO → (K^+^ NIMO^•−#^) → K^+^ + (NIMO^•−^)^#^ → K^+^ + (NIMO–NO)^−^ + NO^•^(4)

The formation of (NIMO–NO)^−^ ion requires the cleavage of C–N and N–O bonds if proceeding from the 4-nitroimidazole ring, or even through other more complex reactions involving the morpholine ring. Nonetheless, the similarity between the DEA resonance profiles and energy positions from NO_2_^–^ and (NIMO–NO)^−^ [3] suggest that these anions have common precursor anion states, and so the reaction leading to the loss of a neutral NO^•^ stems from the 4-nitroimidazole ring. Within the cellular environment, such a reaction is particularly relevant regarding the reactive nature of NO^•^ [42,43,44,45], though producing relevant damage to key biological units including DNA. Yet, if charge transfer is a prevalent mechanism upon the irradiation of biological material rather than single electron attachment, such radical formation from NIMO may certainly not play a relevant role.

The energy feature in Figure 4 at (8.17 ± 0.06) eV yields a vertical electron affinity of (−3.83 ± 0.06) eV. DEA experiments to NIMO assigned the main resonance features at 3.53 and 3.6 eV to CN^−^ and CNO^−^ formation [3]. Given the resonance profiles resemblance between CN^−^ (26 u) and CNO^−^ (42 u) with fragment anions C_3_HN_2_^−^ (65 u) and C_3_H_3_N_2_^−^ (67 u), Meißner et al. [3] concluded that these stem from the imidazole ring. These fragment anions in Table 1 have also been assigned to result from bond breaking within the 4-nitroimidazole ring, in particular from the C–NO_2_ site in the corresponding unimolecular decomposition of the TNI upon electron transfer. Following Meißner et al.’s assignment [3], CN^−^ can proceed from a dissociative mechanism, where the electron transferred to NIMO molecule sits on a $\pi_{CN}^{*}$ antibonding orbital with efficient energy transfer from the CN to the C–CN reaction coordinate. Such is the assertion with the LUMO + 80 character in Appendix A. A close inspection of such MO reveals, apart from the π* character in the 4-nitroimidazole ring, a relevant σ* antibonding character along the C–CN bond. This intramolecular mechanism within the TNI can solely be explained in terms of vibrational predissociation, which can hold as long as the nuclear wave packet survives long enough for the $\pi_{CN}^{*}$ system to diabatically cross with the σ* antibonding state. The morpholine ring fragmentation in Table 1 are assigned to fragment anions from 81 u to 86 u.

The K^+^ energy loss features at (9.55 ± 0.11) eV (Figure 4) with a vertical electron affinity of (−5.21 ± 0.11) eV is tentatively assigned to a Rydberg excitation. The MOs calculation in Appendix A shows the LUMO + 90 with a delocalized spin density resembling a ns Rydberg shape normal to the 4-nitroimidazole ring plane. This feature can be due to an electronic excitation of the series *n*_O_ → ns converging to NIMO ionization energy at 8.15 eV [3]. We tentatively assigned it to (n + 1)s or (n + 2)s due to the large number of states which occur in this high energy region.

## 4. Materials and Methods

### 4.1. The Crossed Molecular Beam Setup in the Lisbon Laboratory

The crossed molecular beam setup used to investigate collisions of neutral potassium (K) atoms with neutral nimorazole (NIMO) molecules has been described in detail elsewhere [18,37]. Briefly, an effusive molecular beam target crossed a primary beam of fast neutral K atoms, and the product anions were analyzed using a reflectron (KORE R-500-6) time-of-flight (r-TOF) mass spectrometer. The K beam was produced in a resonant charge exchange chamber from the interaction of K^+^ ions from a commercial potassium ion source (HeatWave, Watsonville, CA, USA) in the range of 10 up to 400 eV in the lab frame, with gas-phase neutral potassium atoms from an oven source. The TOF anion yield was normalized considering the primary beam current, pressure, and acquisition time. Negative ions formed in the collision region were extracted by a ~380 V/cm pulsed electrostatic field. The typical base pressure in the collision chamber was 6 × 10^−5^ Pa and the working pressure was 2 × 10^−4^ Pa. Mass spectra (resolution m/Δm ≈ 800) were obtained by subtracting background measurements (without the sample) from the sample measurements. Mass calibration was performed on the basis of the well-known anionic species formed after potassium collisions with nitromethane [37] and tetrachloromethane [46].

Post-collision potassium cation (K^+^) energy loss spectra in the forward scattering direction (*θ* ≈ 0°) with the beam’s optical path were recorded in a hemispherical energy loss analyzer. These experiments were not performed in coincidence with TOF mass spectrometry. The analyzer was operated in constant transmission mode to maintain the constant resolution throughout the entire scans. The energy resolution during the experiments was (1.2 ± 0.2) eV, and the energy loss scale was calibrated using the K^+^ beam profile from the potassium ion source serving as the elastic peak.

The molecular sample of NIMO was supplied by Toronto Research Chemicals (North York, Canada) with a stated purity of ≥97%. The solid sample was used as delivered and gently heated up to 343 K using a temperature PID (proportional-integral-derivate controller) unit. A special procedure was carried out to test for any thermal decomposition products within the NIMO beam. Thus, mass spectra were recorded at different temperatures and no differences in the relative peak intensities as a function of temperature were observed.

### 4.2. Theoretical Methods

The electronic structure investigations of the molecular orbitals (MOs) formed in collisions between potassium (K) atoms and NIMO were performed to provide insight into the electron transfer process up to 10 eV. In particular, focus on the analysis of the computed lowest unoccupied molecular orbitals (LUMOs) was considered to assess the nature of the different electronic states that result in the detected negative ions of the current experiments.

The geometries of both NIMO in its ground state and NIMO + K were initially fully optimized at the M06-2X/6-311++g(d,p) level of theory [47]. We started the calculations by positioning the potassium atom at 5 Å from the morpholine and the 4-nitromidazole moieties which is an arbitrary distance. The resulting optimized system upon the collision of the K atoms with NIMO resulted in the structure shown in Figure 5.

All quantum chemical computations were performed with the Gaussian 16 program package [48]. The calculation was carried out in Cartesian coordinates, with no symmetries. All electrons were taken into account for carbon, oxygen, nitrogen, hydrogen and potassium atoms with the 6-311++g(d,p) basis set [49,50]. The natural molecular orbitals for K−NIMO were calculated by Time-Dependent Density Functional Theory (TD-DFT/M06-2X) methods [51].

## 5. Conclusions

The current comprehensive study on the collisions of neutral potassium atoms with neutral NIMO molecules constitutes the most detailed and unique assignment on electron transfer experiments. Here, we combine experimental TOF mass spectrometry and energy loss spectroscopy together with state-of-the art theoretical calculations at different levels of accuracy. The information obtained allowed to investigate the formation of the most representative anions in a wide collision energy range from 10 up to 400 eV in the laboratory frame. Special attention was paid to the non-dissociated parent anion, NIMO^•−^, NO_2_^−^ and other fragment anions stemming from the 4-nitroimidazole and morpholine rings. Although we observed a rich fragmentation pattern related to the multiple bond breaking of both rings, their yields are not significant when compared to NIMO^•–^ at 226 u and NO_2_^−^ at 46 u. The role of K^+^ in the collision complex (K^+^ NIMO^•–#^) seems to be a relevant stabilizing effect at low-energies (E_CM_ < 70 eV), opening up different fragmentation channels despite their lowest BRs, whereas at higher energies, the collision dynamics were mostly dictated by the efficient energy redistribution within the different degrees of freedom of NIMO temporary negative ion, resulting mostly in a stable non-dissociated parent anion and a minor fragmentation yield (~15%).

Potassium cation energy loss spectrum in the forward scattering direction (*θ* ≈ 0°) at 157 eV in the center-of-mass frame (205 eV in the lab frame) shows a dominant feature with a vertical electron affinity of (7.21 ± 0.03) eV assigned to a transition from the NIMO neutral ground state to the repulsive $\sigma_{CN}^{*}$ antibonding state of the TNI, resulting in NO_2_^−^ formation. This energy loss feature shows a threshold at ~5 eV, which is in excellent agreement with the obtained value of 5.26 eV.

We also noted some differences in the rich fragmentation pattern and the anion yields obtained from the present experiments, and those from electron attachment studies. Yet, NIMO is extremely efficient in capturing an extra electron, either from a bound atomic compound via an electron transfer process or from a simple associate electron attachment mechanism. The present results also give relevant information as to the underlying molecular mechanisms governing the electron transfer process, showing the strong ability of this molecule to scavenge an electron, therefore supporting the premise that NIMO is selectively cytotoxic to tumor cells via its reduction as a requirement for accumulation in the cell [52]. Thus, radical formation upon electron capture has been identified to be a pivotal mechanism for the radiosensitization of hypoxic cells by using nitroimidazolic compounds, as is the case of NIMO. The radiosensitization effect is only operative if these chemical compounds are incorporated in cells prior to irradiation of the biological material [52]. Upon redox reaction in cells, the formation of a non-dissociated parent anion (radical anion) is not responsible for cytotoxicity to DNA. The mechanism that seems to be operative in hypoxic cells requires a proton transfer to the reduced chemical compound, yielding a neutral radical species that will bind to DNA, resulting in strand breaks [53]. Additionally, it is interesting to note the relevance of OH^•^ radicals from water radiolysis, which attack DNA at specific sites where nitroimidazolic agents bind to DNA, making these chemical compounds well-attuned to mimicking oxygen in hypoxic tumors. Finally, we believe that the present investigation is relevant to understand how NIMOs accumulating in tumor cells can be used for the further formulation of new radiopharmaceutical compounds.

## Figures and Tables

**Figure 1 molecules-27-04134-f001:**
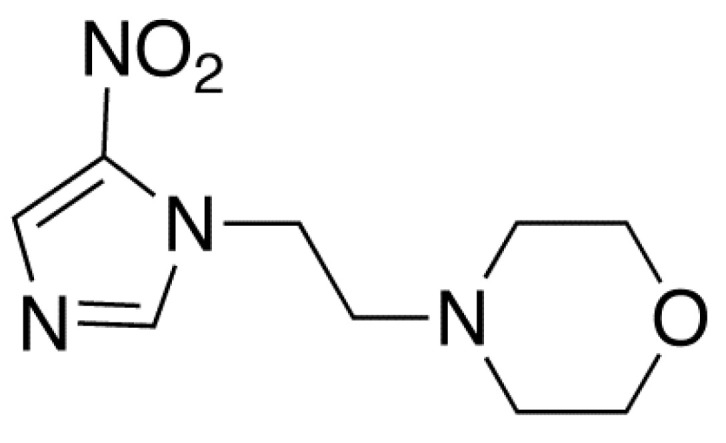
Chemical structure of nimorazole (NIMO), C_9_H_14_N_4_O_3_.

**Figure 2 molecules-27-04134-f002:**
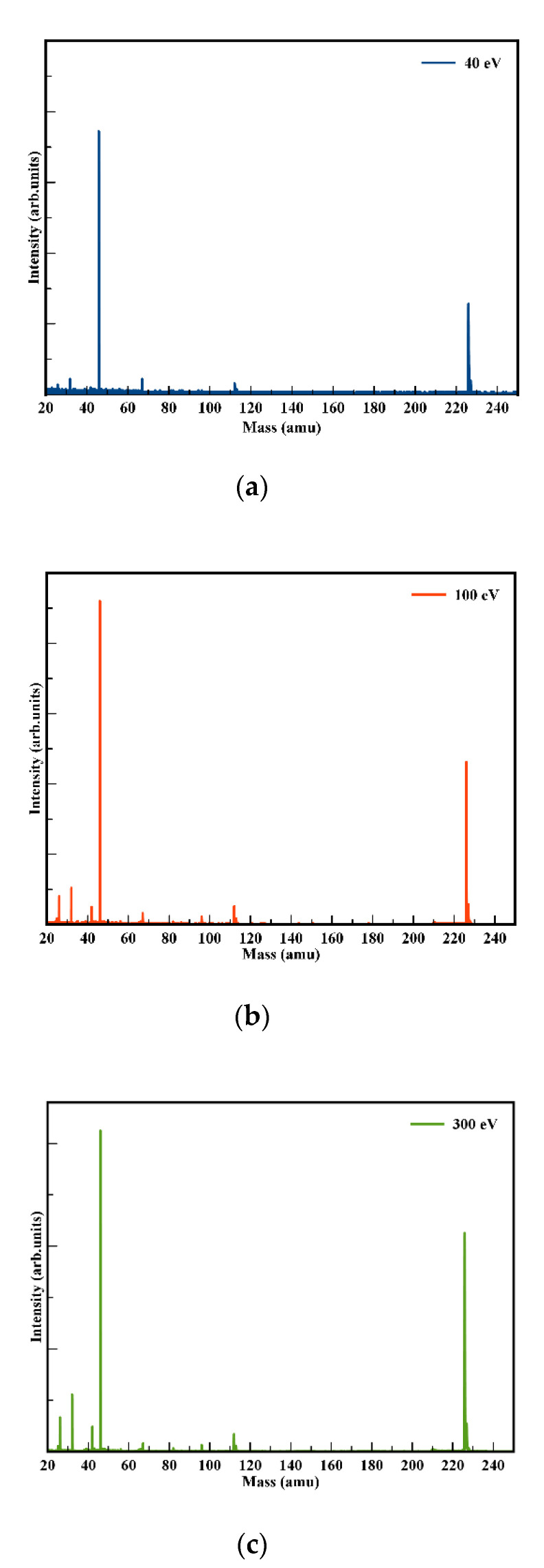
Time-of-flight negative ions mass spectra in neutral potassium collisions with neutral nimorazole at: (**a**) 40 eV lab frame (30.7 eV center-of-mass); (**b**) 100 eV lab frame (76.7 eV center-of-mass); (**c**) 300 eV lab frame (230.2 eV center-of-mass). See text for details and Table 1 for assignments.

**Figure 3 molecules-27-04134-f003:**
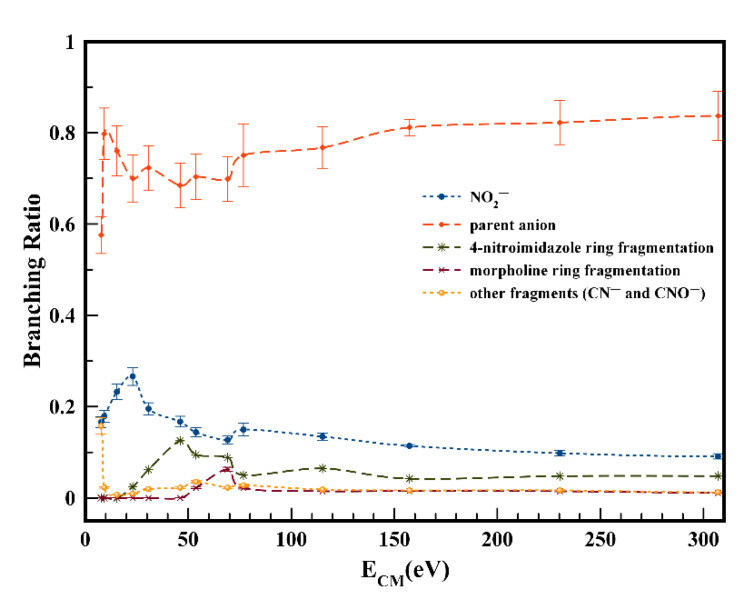
NIMO branching ratios (fragment anion yield/total anion yield) of the main anions formed as a function of the collision energy in the center-of-mass frame. Error bars are related to the experimental uncertainty associated with the ion yields. The lines were added just to guide the eye. See text for details.

**Figure 4 molecules-27-04134-f004:**
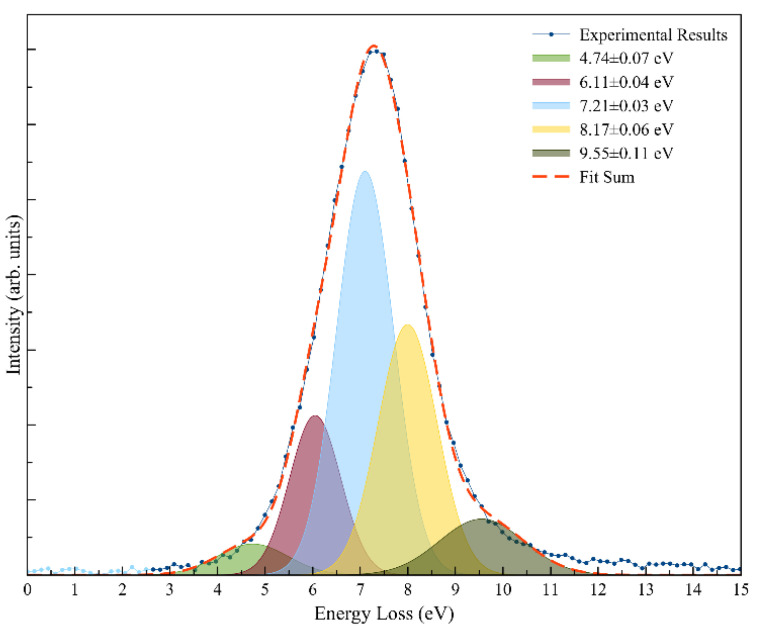
Post-collision potassium cation (K^+^) energy loss spectrum at 157 eV in the center-of-mass frame (205 eV in the lab frame) in the forward scattering direction (*θ* ≈ 0°). The peaks’ uncertainty result from the Gaussian fitting procedure. See text for details.

**Figure 5 molecules-27-04134-f005:**
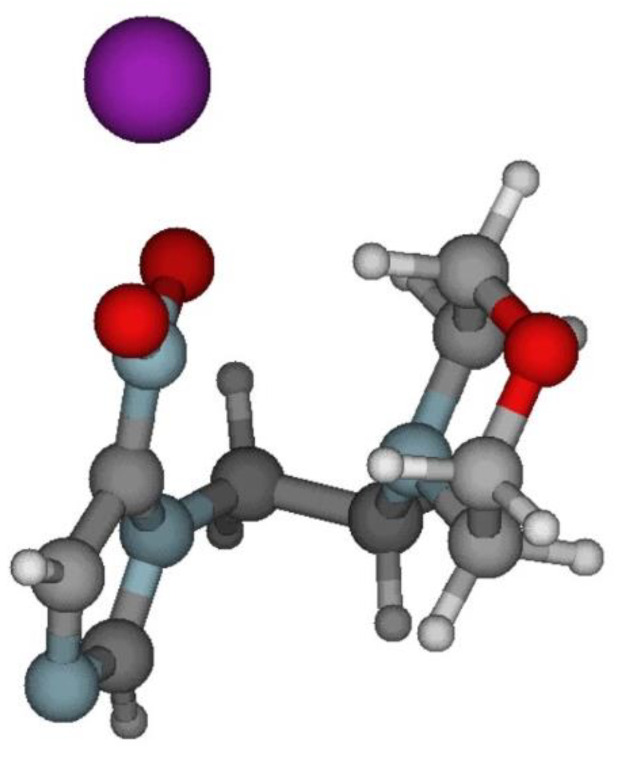
Optimized molecular structure of nimorazole, and orientation of the K + NIMO collisional system K − O ≈ 5.1 Å. K: yellow, O: red, C: grey, N: light blue, and H: white. See text for details.

**Figure 6 molecules-27-04134-f006:**
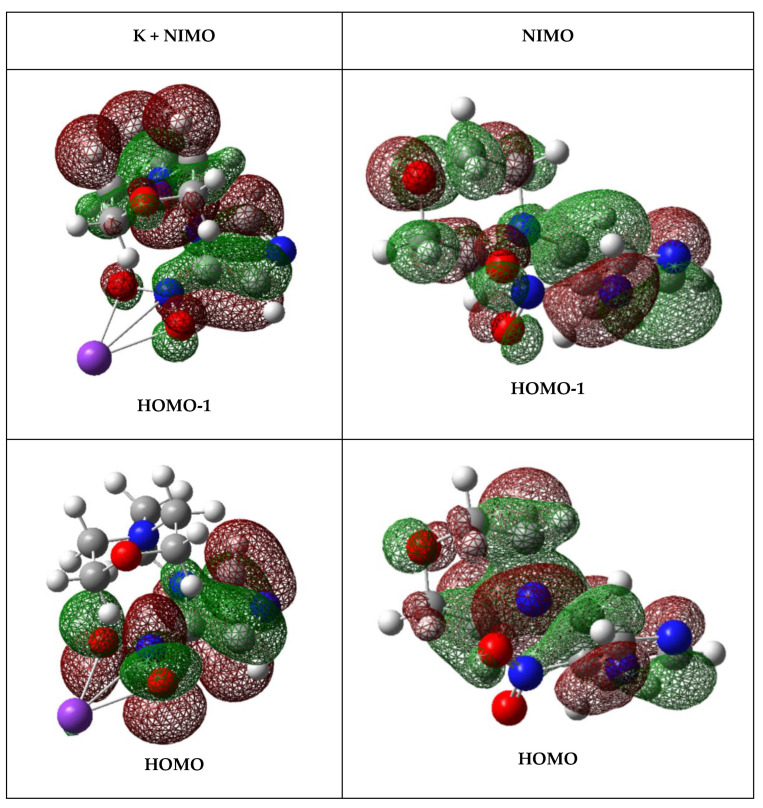
Calculated highest occupied molecular orbitals (HOMO-1 and HOMO) and lowest unoccupied molecular orbital (LUMO + 50) for K + NIMO and NIMO (K: purple, C: grey, N: blue, O: red, and H: white). The straight lines between the K atom and the –NO_2_ end in the 4-nitroimidazole ring are just to indicate the spatial mutual position. For a complete set of MOs see Appendix A.

**Table 1 molecules-27-04134-t001:** Proposed assignment of negative ions formed in electron transfer in potassium collisions with nimorazole (NIMO) compared with electron attachment experiments [2,3].

Mass (u)	Mass (u) [2,3]	Assignment	Major Source of Fragmentation
26, 27	26	CN^−^, ^13^CN^−^	4-nitromidazole ring
30	−	NO^−^	4-nitromidazole ring
38	−	C_2_N^−^/C_3_H_2_^−^	
39	−	C_2_HN^−^/C_3_H_3_^−^	
40	−	C_2_H_2_N^−^	
41	−	C_2_H_3_N^−^/CHN_2_^−^	
42	42	CNO^−^	4-nitromidazole ring
43	−	CHNO^−^	
46, 47, 48	46, 48	NO_2_^−^, ^15^NO_2_^−^, ^14^N^18^OO^−^	4-nitromidazole ring
52	−	C_3_H_2_N^−^	
56	−	C_2_H_2_NO^−^/C_3_H_6_N^−^	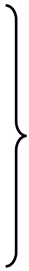	4-nitromidazole ring
64	−	C_3_N_2_^−^
65	65	C_3_HN_2_^−^
66	−	C_3_H_2_N_2_^−^
67	67	C_3_H_3_N_2_^−^
68	−	C_3_H_4_N_2_^−^
69	−	C_3_H_5_N_2_^−^
79	−	C_4_H_3_N_2_^−^	
80	−	C_4_H_2_NO^−^	
81	−	C_3_HN_2_O^−^/C_4_H_3_NO^−^	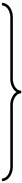	morpholine ring
82	82	C_3_H_2_N_2_O^−^/C_4_H_4_NO^−^
83	83	C_3_H_3_N_2_O^−^/C_4_H_5_NO^−^
84	−	C_3_H_4_N_2_O^−^/C_4_H_6_NO^−^
86	−	C_4_H_8_NO^−^
93	−	C_4_HN_2_O^−^	
94	−	C_3_N_3_O^−^	
95	−	C_4_H_3_N_2_O^−^	
96	−	C_3_H_2_N_3_O^−^	
97	97	C_3_H_3_N_3_O^−^	
108	−	C_4_H_2_N_3_O^−^	
109	−	C_4_HN_2_O_2_^−^	
111	−	C_4_H_3_N_2_O_2_^−^	
112	112	C_3_H_2_N_3_O_2_^−^	
113	−	C_4_H_5_N_2_O_2_^−^	
114	−	C_3_H_4_N_3_O_2_^−^	
126	−	C_4_H_4_N_3_O_2_^−^	
134	−	C_5_N_3_O_2_^−^	
138	−	C_5_H_4_N_3_O_2_^−^	
139	−	C_4_H_3_N_4_O_2_^−^	
−	178	(NIMO–NO_2_–2H)^−^	
196	196	(NIMO–NO)^−^	
−	208	(NIMO–H_2_O)^−^	
−	209	(NIMO–OH)^−^	
210	210	(NIMO–O)^−^	
225	225	(NIMO–H)^−^	
226, 227, 228, 229	226	NIMO^•–^ and its isotopes	

**Table 2 molecules-27-04134-t002:** Assignment of different features from the Gaussian fitting to K^+^ energy loss spectrum from potassium collisions with NIMO at 205 eV lab frame energy (157 eV in the center-of-mass frame). VEA (Vertical Electron Affinity), VE (Vertical Energy), EA (Electron Attachment) *.

K^+^ Energy Loss Feature (eV)	VEA (eV)	Calculated VE of MO (eV)	Assignment	EA Resonances [2,3]
4.74 ± 0.07	−0.40 ± 0.07	4.81	LUMO + 50	0.41
6.11 ± 0.04	−1.77 ± 0.04	5.69	LUMO + 60	1.49, 1.93
7.21 ± 0.03	−2.87 ± 0.03	7.01	LUMO + 70	2.97
8.17 ± 0.06	−3.83 ± 0.06	8.28	LUMO + 80	3.53, 3.6
9.55 ± 0.11	−5.21 ± 0.11	9.17	LUMO + 90	–

* The uncertainties result from the Gaussian fitting procedure used.

## Data Availability

Not applicable.

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
