# Peer review of "Bound Electron Enhanced Radiosensitisation of Nimorazole upon Charge Transfer"

_molecules, 2022, doi:10.3390/molecules27134134_

Round 1
Reviewer 1 Report
Review of the ms "Bound electron enhanced radiosensitisation of nimorazole upon charge transfer" by Kumar et al.
This manuscript presents a study of the collisions of neutral potassium atoms with Nimorazole, covering a relatively large translational energy range. For such a cross molecular beams setup and TOF mass spectrometry are used. Theoretical calculations for the rationalization of some of the results were also reported.
I consider the document is well presented, and the results find support through the text. However, there are some issues requiring attention before the publication at Molecules.
Specific comments.
-Is there any motivation for the selected collision conditions? i.e. why was potassium selected as a collision partner, and why such a translational energy range was selected?
-Similarly, is there any support for the selected level of theory in the theoretical calculations? Orbital's energy strongly depends upon the selected theoretical strategy, such a selection requires some support. Ex. results reported in previous works or exploring/comparing spectroscopical properties.
-How many and which electronic states were considered in the molecular structure calculations? (lines 379-383)
-Is the distance of 5A arbitrarily selected? (lines 386-387)
-I suggest using only one color representing the K atom, instead of two (yellow and purple) in the Figures.
Author Response
We are grateful to the reviewer for his/her perceptive and constructive comments and a reply is included.
-Is there any motivation for the selected collision conditions? i.e. why was potassium selected as a collision partner, and why such a translational energy range was selected?
Author's reply: the energy range used was selected to allow determining thresholds of formation for a set of selected anions as well as to obtain relevant information on the collision dynamics. Note that the role of the collision complex and the stabilization effect induced by the K+ ion after the collision process, can only be achieved by covering a wide collision energy range. This is in assertion with the branching rations trending in Figure 3 and the rationale put forward in the discussion section. Moreover, the energy loss data is also in assertion with the need to explore at higher collision energies the role of vertical transitions into the negative ion states attained in such processes. Potassium has been thoroughly used by our research group over the last 15 years or so, simply because we had it available, although we might have decided for another alkali such as sodium or even caesium. The main rationale of an alkali choice is due to its low ionisation energy (4.34 eV) making this well-attuned to electron transfer experiments.
-Similarly, is there any support for the selected level of theory in the theoretical calculations? Orbital's energy strongly depends upon the selected theoretical strategy, such a selection requires some support. Ex. results reported in previous works or exploring/comparing spectroscopical properties.
Author's reply: yes, that is what some of the manuscript is about. Given the novelty of the present work, there is certainly no comparative studies to look at. However, the information gained from the energy loss data, the related vertical electron affinities of the attained electronic states, are well supported by the present calculations showing the relevant electronic contributions yielding selected anions being formed. The calculated energies for each molecular orbital relevant for that electronic state are in very good agreement with the experimental findings (see Table 2) and are a measure of the quality of the theoretical level of accuracy given the related information obtained via electron attachment and dissociative electron attachment experiments on the accessible resonances.
-How many and which electronic states were considered in the molecular structure calculations? (lines 379-383)
Author's reply: as far as excited states are concerned, the current study does not account for that. Yet, we have only computed all molecular orbitals that can generate electronic excited states, i.e., by replacing occupied MOs with virtual MOs (LUMOs). Moreover, the methodology employed followed the information from the electron attachment investigation in ref. [3]. Thus, we have initiated the calculations from that geometry of NIMO (charge = 0 and total spin S = 0) and focused on getting an electronically stable wave function.
-Is the distance of 5A arbitrarily selected? (lines 386-387)
Author's reply: the 5Å distance was not arbitrary. It is obtained when K + NIMO was globally optimised, i.e., when all coordinates are relaxed.
-I suggest using only one color representing the K atom, instead of two (yellow and purple) in the Figures.
Author's reply: changed accordingly.
Reviewer 2 Report
Nimorazole is widely used as an cell radiosensitizer. Primary ionizing radiation usually produce significant amount of low-energy secondary electrons and a lot of other secondary species, including positive ions. Thus, the details of charged particles interactions with the studied target molecule is important. The work reports deduction due to electron transfer in collisions with neutral potassium atoms. Experimental results collected with time of flight mass spectrometer are suppoted by theoretical calculations performed with Gaussian code. Results are well presented and discussed. They can be very useful in detailed studies and simulations of ionizing radiation interaction with the radiosensitized cells. I recommend this work for publication in "Molecules" MDPI journal.
Author Response
We are grateful to the reviewer for his/her perceptive and constructive comments, and given that there are no suggested changes, no further action will be taken.